# EgSPEECHLESS Responses to Salt Stress by Regulating Stomatal Development in Oil Palm

**DOI:** 10.3390/ijms23094659

**Published:** 2022-04-22

**Authors:** Zhuojun Song, Le Wang, Chongcheong Lai, May Lee, Zituo Yang, Genhua Yue

**Affiliations:** 1Molecular Population Genetics and Breeding Group, Temasek Life Sciences Laboratory, 1 Research Link, National University of Singapore, Singapore 117604, Singapore; zhuojun@tll.org.sg (Z.S.); wangle@tll.org.sg (L.W.); astrolai@gmail.com (C.L.); maylee@tll.org.sg (M.L.); ituo@tll.org.sg (Z.Y.); 2Department of Biological Sciences, National University of Singapore, 14 Science Drive 4, Singapore 117543, Singapore

**Keywords:** oil palm, salt stress, stomata, signalling, SPEECHLESS

## Abstract

Oil palm is the most productive oil producing plant. Salt stress leads to growth damage and a decrease in yield of oil palm. However, the physiological responses of oil palm to salt stress and their underlying mechanisms are not clear. RNA-Seq was conducted on control and leaf samples from young palms challenged under three levels of salts (100, 250, and 500 mM NaCl) for 14 days. All three levels of salt stress activated EgSPCH expression and increased stomatal density of oil palm. Around 41% of differential expressed genes (DEGs) were putative EgSPCH binding target and were involved in multiple bioprocesses related to salt response. Overexpression of EgSPCH in Arabidopsis increased the stomatal production and lowered the salt tolerance. These data indicate that, in oil palm, salt activates EgSPCH to generate more stomata in response to salt stress, which differs from herbaceous plants. Our results might mirror the difference of salt-induced stomatal development between ligneous and herbaceous crops.

## 1. Introduction

Oil palm (*Elaeis guineensis*, Jacq.) produces the highest yield of plant oils [1]. Due to the negative effects of oil palm expansion, such as deforestation and decreasing biodiversity, sustainable plantation and management are crucial to increase oil production and minimize damage to the environment [2]. Oil palm is cultivated in tropical areas of Asia, Africa, and America [1], where many coastal soils of those areas are salinized due to tidal waters [3]. The fresh fruit bunch (FFB) yields of oil palm are dramatically decreased on saline soils [3]. Therefore, the genetic improvement by selecting salt-tolerant oil palm varieties is important for sustainable palm oil production [1]. However, not much is known about the molecular mechanism underlying salt tolerance in oil palm.

Over the past decade, the molecular mechanisms of salt tolerance have been largely studied in *Arabidopsis* and agronomic plant species, such as rice [4,5]. Salt stress can directly alter the biological compounds physically or chemically in plant cells, which activates cellular response [5]. Furthermore, salt stress leads to ionic stress, secondary stresses, osmotic stress, and oxidative stress, thereby triggering multiple complex signaling pathways [6]. The leucine-rich repeat extensins (LRX)-Raf-like kinase (RALF)-FERONIA (FER) module is important for cell wall integrity and cell wall-associated biological processes [7]. In plants, high salinity disrupts the cross-link between pectin and LRXs and the interaction between LRXs and RALFs, resulting in cell bursting during growth under salt stress [8]. Salt stress triggers the cytosolic Ca^2+^ signal, which activates the Na^+^ homeostasis required by the Salt Overly Sensitive (SOS) signaling pathway. Ultimately, H^+^-ATPase is activated and Na^+^ is exported via Na^+^/H^+^ exchanger driven by H^+^-ATPase [4,5]. Many other genes are also important in the ionic stress signaling pathway. They repress the salt sensory system, limit the salt absorption and transportation in plants, regulate root and leaf development, and adjust the ionic balance of cells to raise the salt tolerance [9,10]. Transcription factors (TFs) play key roles in the salt stress tolerance of plants. They are differentially expressed during salt stress, which consequently regulate the transcription of various downstream genes that are involved in salt tolerance [11]. The most well-known salt tolerances associated with TFs include basic leucine zipper (bZIP), basic helix-loop-helix (bHLH), MYB, WRKY, APETALA2, and NAC [5,11,12]. Among the TFs, a bHLH transcription factor SPEECHLESS (SPCH) serves as a master nuclear localized regulator of cell development in response to environmental changes [13]. SPCH, MUTE, and FAMA are three bHLH homologs in regulating stomatal development [14]. Among them, SPCH binds to ~4.5% of genes in Arabidopsis, including key genes in abiotic stress and the hormonal stress signaling pathway [13]. The function of SPCH in stomatal initiation is conserved in both dicots and monocots [15,16]. Under salt stress, the expression of SPCH was repressed by upstream transcriptional factors and the mitogen-activated protein kinase (MAPK) signaling pathway, resulting in the reduction of stomatal production to avoid water loss [17]. Although these studies provide novel knowledge and new insights into the regulatory networks of salt tolerance, the complexity of salt resistance, the genetic divergence of different species, and the diversity of environments make it difficult to understand the particular mechanisms of other plants in response to salt stress [12].

Stomata are minute openings found in the epidermis of the plants, which control CO_2_ intake for photosynthesis and regulate water loss. Stomata consist of pairs of guard cells, which are required for stomatal movement [18]. The ATP driven proton pumps in guard cells are key elements for stomatal movement, which are highly sensitive to various environmental changes [18]. Salt stress is also considered to be a physiological drought to plants [19]. Numerous studies have shown that the reduction and closure of stomata in herbaceous plants are beneficial to minimize residual transpiration and improve water use efficiency and Reactive oxygen species (ROS) scavenging [20,21,22]. In contrast, a few works in woody plants have shown an increase in stomatal density under salt stress. Salt tolerant trees exhibit higher stomatal density and smaller stomatal size [23]. 

Few studies show the physiological and proteomic changes of palms in response to salt stress. In oil palm seedlings subjected to salt stress, there was an increase in Na^+^ and proline content, and the cell membrane was injured in samples treated with the highest salinity at 200 mM NaCl. On the contrary, photosynthetic and growth rate were reduced [24]. A proteome study of date palm suggests that ATP synthase and Ribulose-1,5-bisphosphate carboxylase-oxygenase (RubisCO) activase are significantly changed during salt stress [25], indicating the importance of biosynthesis for salt tolerance. These studies show the physiological responses of palms under salt stress. However, the cellular level response and the molecular mechanisms of the salt tolerance of palms are still unknown. 

The purpose of this study is to investigate the salt response of oil palm on the cellular level and identify the critical regulators and signaling pathways involved in salt-tolerance. Herein, we found that oil palm exhibits aa diverse biological strategy in response to different levels of salt stress. Furthermore, we found that salt stress activates *EgSPCH* expression, which leads to the increase of stomata. By comparing with the ChIP-seq dataset of AtSPCH [13], EgSPCH putatively regulates the expression of 41% of the differential expressed genes (DEGs). Our study revealed that the different salt response of stomatal development in oil palm and other herbaceous crops may rely on the salt-induced upstream regulation of *SPCH*.

## 2. Results

### 2.1. Morphological and Physiological Responses to Salt Tolerance

After 14 days of drought stress, common plant stress responses, including leaf tip necrosis, leaf yellowing, and wilting, were observed in all the salt treated samples in varying degrees (Figure 1a). With increasing salt concentration, the above responses of leaf and roots were enhanced (Figure 1a,c). Interestingly, salt was emitted and crystalized on leaf epidermal of oil palms treated with high concentration of salt at 250 mM and 500 mM (Figure 1c, Appendix A). To investigate the effect of salt stress on later growth of plants, rescue assay was performed by giving all the samples 150 mL of water daily for another 14 days. Plants that were given 2-weeks of salt treatment with 100 mM NaCl and 250 mM NaCl survived after rescue assay. However, 500 mM NaCl was lethal to oil palm seedlings (Figure 1b).

### 2.2. The Effect of Salt Stress on the Stomatal Development of Oil Palm 

To understand how stomata of oil palm respond to salt stress, the stomatal development of samples from mock and salt stress groups were monitored (Figure 2a). In salt-treated groups, the stomatal index and density were higher than that in the control group, while there was no difference among these salt stress groups (Figure 2b,c). In addition, the stomatal production among each salt stress group was unchanged (Figure 2b,c). To test whether the increase of stomata density was caused by possible increase of the total cell number, the total epidermal cell number and the stomatal index were analyzed. The total epidermal cell numbers were comparable between the mock and salt treatment group (Appendix A). These data indicate that salt-induced osmotic stress activates the stomatal production of oil palm. 

### 2.3. The DEGs in Response to Different Levels of Salt Stress

Average cleaned RNA-seq reads of 46.2 (±2.1), 35.0 (±3.1), 59.4 (±1.4), and 35.9 (±3.8) million were obtained from the mock, 100 mM, 250 mM, and 500 mM NaCl groups, respectively (Appendix A). A total of 363, 242, and 433 DEGs were identified from salt stress groups (100, 250, and 500 mM NaCl, Figure 3). In detail, there were 86 down-regulated and 277 up-regulated DEGs in the 100 mM NaCl group, 155 down-regulated and 87 up-regulated DEGs in the 250 mM NaCl group, and 249 down-regulated and 184 up-regulated DEGs in the 500 mM NaCl group (Figure 3A). Three DEGs *EgSPCH (XP_010915684.1)*, *EgPAT1 (LOC105046259)*, and *EgRPS3 (LOC105040725)* were up-regulated in all the salt treatment groups (Figure 3A). In addition, *EgFAMA* (XP_010909228.1), another key stomatal regulator and a homolog gene of *EgSPCH*, was also up-regulated in the 500 mM NaCl group (Appendix A). The Log_2_FC of the other homolog gene of *EgSPCH*–*EgMUTE* (XP_010931955) was 1.68 (*p*-value = 0.5). Although *EgMUTE* was not in the DEG list, it might also be up-regulated. These data suggested that the increase of stomatal production of oil palm in response to salt stress might be a consequence of the up-regulation of the three bHLH stomatal regulators *EgSPCH*, *EgMUTE*, and *EgFAMA*.

### 2.4. The Putative Binding Targets of EgSPCH

The alignment of our DEGs to the ChIP-seq dataset of AtSPCH targets in *Arabidopsis* (Lau et al., 2014) showed that ~40.9% of DEGs (with 60.1% and 38.6% of up- and down- regulated genes, respectively) were putative targets of EgSPCH (Figure 4a). Gene Ontology (GO) terms of these genes involved salt-tolerance, including hormonal and abiotic stress stimulus, developmental processes, organic compound biogenesis, and metabolic processes (Figure 5c). Signaling network analysis by KEGG revealed the regulatory network of putative EgSPCH binding DEGs (Appendix A). For example, the upstream flavonoid synthesis regulator EgCYP75B1 (XP_010932510.1), EgTT6 (XP_010934537.1), and EgTT4 (XP_010942344.1) were involved in the flavonoid accumulation during salt stress (Appendix A). The photosystem I light harvesting complex genes (XP_010913989.1, XP_010939818.1, XP_010916097.1) might affect the photosynthesis (Appendix A). Numerous genes were involved in glycolysis (Appendix A). The key genes of the auxin signaling pathway–EgARFs and jasmone signaling pathway–EgJAZs were also involved during salt stress (Appendix A). 

Among the DEGs, EgSPCH putatively binds to bHLH, MYB, C2H2, NAC, bZIP, and many other transcription factors (Figure 5b, Appendix A). They were involved in multiple biological processes of environmental response. The high percentage of EgSPCH targets among DEGs suggests that EgSPCH may be a key transcriptional switch for salt tolerance of oil palm.

### 2.5. Overexpression of Oil Palm SPCH Facilitates Stomatal Development and Decrease Salt Tolerance in Arabidopsis

The phylogenic tree was clustered into four groups, and each gene of a different species formed one cluster (Figure 4d). SPCH and its two homologs, MUTE and FAMA, were far from the outgroup HAP2, indicating a relatively conserved evolution of these three genes (Figure 4d). Nevertheless, SPCH showed a genetic divergence between ligneous and herbaceous plants. In contrast, MUTE and FAMA were likely less conserved where the OsjFAMA is closer to the MUTE gene groups (Figure 4d).

Two overexpression transgenic lines of the *EgSPCH* (*35S:EgSPCH-YFP#1* and *35S:EgSPCH-YFP#2*) were used to examine the function of EgSPCH in stomatal production and salt tolerance. The transformation was validated by fluorescence microscopy and PCR (Figure 5a, Appendix A). EgSPCH was localized in the nucleus of epidermal cells of 3 dpg abaxial cotyledons, including meristemoid and stomatal lineage ground cell (SLGC) (Figure 5a). There were more small cells and pavement cells in the 3 dpg *35S:EgSPCH-YFP* seedlings than that of Col-0 (Figure 5c). In addition, more stomata were found in the *35S:EgSPCH-YFP* than in the Col-0 at 7 dpg (Figure 5b). These data indicated that the introduction of *EgSPCH* increased the small cell number of early-stage cotyledon, which might lead those cells to a higher transition to stomata. 

The result of the salinity assay showed that the *35S:EgSPCH-YFP* plants had a slightly lower survival rate after 7 days of salt stress. However, after 14 days of salt treatment, ~40% of the Col-0 seedlings survived, while only ~23% of the *35S:EgSPCH-YFP* plants lived (Figure 6a,b). The 14 dpg *35S:EgSPCH-YFP* plants showed higher stomatal production than Col-0, while both Col-0 and *35S:EgSPCH-YFP* exhibited decreased stomata after 7 days of salt treatment (Figure 6c,d). These results indicate the functional similarity of EgSPCH and AtSPCH in facilitating stomatal development and the reduction of salt tolerance.

## 3. Discussion

### 3.1. The Salt Response of Oil Palm

Seven days’ treatment of 200 mM and 150 mM NaCl are half-lethal to rice/wheat [26,27] and *Arabidopsis* [28], respectively. Although oil palm is not a halophyte, pot-grown oil palm seedlings could survive under 250 mM NaCl for at least 14 days with no serious defects, and those defects could be recovered (Figure 1). Studies have shown that they could also survive after long-term (2 months) drought [29]. This suggests that oil palm seedlings have a higher osmotic tolerance compared to staple herbaceous crops.

Studies have shown that the reduction of stomata is beneficial for herbaceous plants to improve water use efficacy in response to osmotic stress [20,21,22]. It was interesting but unexpected to find that salt stress increases stomatal production of oil palm (Figure 2). This result was consistent with the study in another ligneous plant white poplar, where salt increases stomatal production and the salt tolerant poplar 14P11 exhibits a higher stomatal density than that of salt susceptive trees [23]. A recent study shows that stomata is strongly co-varied with the xylem across a large scale of ligneous seedlings. Sufficient stomatal density is critical for water delivery through xylem vessels by the stomatal driven force via evaporation [30]. The above study indicates a specific stomatal distribution and water transport network of ligneous plants, which might play a role in salt tolerance of ligneous plants. The salt tolerance of ligneous plants become stronger from the seedling stage to the juvenile stage. Under salt stress, ligneous plants decrease the vessel lumen thickness to reduce cell cavitation in xylem [31]. In our study, salt was found on the leaf epidermis of oil palm under high salt stress (Figure 1, Appendix A). Interestingly, in halophytes, stomata are channels for salt discharge [32]. It would be interesting to use definitive measurements, such as leaf ion content and gas exchange, to test whether the discharged salt was NaCl and whether it was discharged via the transpiration stream from the stomata of oil palm. The stomata of some halophytes function well in high salinity that would kill most other plants [33]. ABA content remains constant in their leaves, whilst polyphenols, specifically flavanols, accumulated much faster and maintained a higher content level in guard cells of halophytes than in the glycophytes [34]. It would be helpful to monitor the dynamic accumulation of flavonols to understand the physiological stomatal changes of ligneous and herbaceous plants. Taken together, it was possible that the specific co-variation of stomata–xylem of ligneous plant may lead to a different salt response of stomata. The increase of stomata might be required for ligneous plants to reduce the osmotic potential in plant cells by discharging salt and facilitate water transport via the xylem, so that oil palm and white poplar do not need to sacrifice too much of their gas exchange efficiency to compensate their stomata–dependent water use efficiency, such as herbaceous plants.

### 3.2. The Evolution of EgSPCH and Its Putative Binding DEGs

SPCH and its two bHLH homologs, MUTE and FAMA, are essential for stomatal development [14]. Although *EgMUTE* was not a DEG in our study, it was also up regulated. In addition, *EgFAMA* was a DEG in the 500 mM NaCl group. This might be a result of the up-regulation of EgSPCH because, in *Arabidopsis*, SPCH binds to *MUTE*/*FAMA* and activates their expression [13]. Therefore, in oil palm, salt induces both the early initiation and the later cell transition of stomata via activating the expression of *EgSPCH*, *EgMUTE*, and *EgFAMA*. A phylogenic result showed that these three were relatively conserved across different species (Figure 4d). However, interspecific genetic divergence between ligneous and herbaceous species still existed, where *EgSPCH* and *PaSPCH* exhibited the highest genetic similarity (Figure 4d). SPCH is not only a stomatal initiator but is also a master transcriptional factor which binds to thousands of genes in *Arabidopsis*, including key genes involved in the abiotic stress signaling pathway. The transcription of SPCH is also regulated by other transcriptional factors [13]. The ChIP-seq is difficult for oil palm; therefore, our DEGs from RNA-seq were aligned to the ChIP-seq database of AtSPCH [13]. For EgSPCH putatively bound to ~41% of DEGs, a considerable number were involved in salt response-related signaling pathways, such as hormonal, metabolic, and developmental pathways (Figure 4a–c, Appendix A). Nevertheless, the binding motif of AtSPCH-‘CDCGTG’ [13] was not found in either EgSPCH or PaSPCH, suggesting a different binding site and affinity of EgSPCH to its targets. The actual upstream regulators and downstream targets of EgSPCH might be different with AtSPCH. The different genetic structure, including transcriptional binding motifs between EgSPCH and AtSPCH, suggested that their up- and downstream regulatory networks might be different. We provided genome-wide gene candidates that might be regulated by EgSPCH in response to osmotic stresses. However, further molecular studies, such as yeast one hybrid, luciferase reporter assay, and functional verification, in model plants would be important to identify the binding affinity and molecular function of these genes.

### 3.3. The Upregulation of EgSPCH, Rather Than Its Function, Is the Key to Understanding the Salt Response of Stomatal Development in Oil Palm

Our data showed the similar subcellular localization of EgSPCH (Figure 5a) and AtSPCH in the nuclear of stomatal lineage cells.The function of EgSPCH in stomatal initiation and salt tolerance was also comparable with other plants, including monocot plants, where SPCH increases the stomatal production but salt represses the expression of *SPCH* [4,16]. These data indicate that the activation of *EgSPCH* by salt in oil palm is not monocot specific. Importantly, the genetic difference of *SPCH* between ligneous and herbaceous plants did not lead to the functional divergence of them in stomatal development and salt response. Taken together, the different response of SPCH to salt in oil palm and *Arabidopsis* is likely the result of different upstream regulation of SPCH due to the different genetic structure including binding motifs. Moreover, to understand the salt response of EgSPCH at the protein level, we performed the MPK3/6 Western blot assay. However, the p44/42 antibody (cell signaling technology, USA), which works well in *Arabidopsis*, did not work in oil palm (data not shown). It would be valuable to test the stomatal behavior and the SPCH expression at transcriptome and protein levels in more ligneous plants to understand whether the regulatory mechanism of SPCH in response to salt was reserved in ligneous plants.

## 4. Materials and Methods 

### 4.1. Plant Materials and Salt Treatment

Sixteen two-year-old oil palm seedlings of similar sizes were planted in 20 cm diameter pots and were placed in a greenhouse with tropical temperatures and natural photoperiods. The seedlings were divided into four groups (4 seedlings for each group): The mock group (control group) was watered daily with 150 mL sterilized water, while the salt stress groups (100 mM, 250 mM, and 500 mM NaCl group) were watered daily with equal volumes of 100 mM, 250 mM, and 500 mM NaCl diluted by sterilized water, respectively. This is to simulate the condition of the increasing soil salinity caused by mineral weathering or ocean withdrawal. After 14 days of the salt stress challenge, the young rosette leaves of similar size in each group were collected. 

Col-0 and transgenic *35S:EgSPCH-YFP Arabidopsis* seeds were sterilized and grown on ½ MS plates (0.5 g/L MES, 2.2 g/L Murashige and Skoog salts, 1% [*w*/*v*] sucrose, and 0.8% [*w*/*v*] agar, pH 5.6) and kept at 4 °C in darkness for 3 days. Plants were grown in a growth chamber at 22 °C with 60% relative humidity under long-day conditions (16 h light/8 h dark) at a light intensity of 70 μmol m^−2^ s^−1^. At 7 dpg, 40 well-grown seedlings were transferred to either new ½ MS plates (Control) or ½ MS + 100 mM NaCl plates.

### 4.2. Plasmid Construction and Plant Transformation

To generate *35S:EgSPCH-YFP*, the full length CDS sequence of *EgSPCH* was amplified and cloned into pENTR/D-TOPO (Thermo Fisher, Waltham, MA, USA), after which the entry clone was recombined into the destination vector pGWB541 [35] via LR recombination using Gateway LR Clonase II (Thermo Fisher, MA, USA). The primers (*EgSPCH*cds-F/R) used for plasmid construction are listed in Appendix A. Transgenic plants were generated in the Col-0 background through *Agrobacterium tumefaciens*-mediated transformation [36] and selected by hygromycin on ½ MS plates.

### 4.3. RNA Extraction and Sequencing

Total RNA from oil palm leaves of three biological replicates of the control (Mock group) and salt treated samples (100 mM, 250 mM, and 500 mM NaCl) was extracted using RNeasy Plant Mini Kit (Qiagen, Hilden, Germany). RNA quality and quantity assessment, RNA-seq library preparation, library quality control, and library quantification were performed using a previously described method [37]. The libraries were sequenced with an Illumina NextSeq500 (Illumina, CA, USA). 

### 4.4. Measurement of Stomatal Production 

Small slices from each young rosette leaves collected after 14-day mock or salt treatment were immediately stained with propidium iodide (PI, Molecular Probes, P3566; 0.1 mg/mL) for cell integrity fluorescence microscopy, and images were captured at 20× on a ZEISS Axioscan 7. For quantification of stomatal density and aperture, fresh leaf slices were first cleared in fixing buffer (7:1 ethanol: acetic acid) for 8 h and were mounted in clearing buffer (8:2:1 chloral hydrate: water: glycerol). Differential contrast interference (DIC) images of the abaxial epidermis of young leaf slices were captured at 20× on a Leica DM2500 microscope. More than 20 slices were examined per test. Stomatal density and index were measured by ImageJ with its built-in tools.

### 4.5. Bioinformatical Analysis

Adaptor filtering and cleaning of raw sequencing reads were carried out using SeqKit [38]. Cleaned reads were aligned and mapped to the oil palm reference genome [39,40] using STAR [41]. The expression level of each gene was counted using HTSeq-count against gene annotations [42] and the relative expression of each gene was normalized using DESeq2 [43]. Transcripts with more than two times fold change (FC) value and a significance value less than 0.05 were considered as differentially expressed genes, between mock and salt treatment groups.

The protein sequences of SPCH and its two bHLH homologs, MUTE and FAMA, of three herbaceous plants, *Arabidopsis (At), Oryza sativa L. ssp. Japonica (Osj),* and *Glycine max (Gm)* and three ligneous plants *Elaeis guineensis* (*Eg*), *Eucalyptus grandis* (*Eug*), and *Populus alba L* (*Pa*) were aligned by ClustalW [44]. The protein HAP2 was used as the outgroup reference. The gene bank number of above protein sequences are: AtSPCH - ABI26170.1, AtMUTE-OAP03215.1, AtFAMA-NP_189056.2, AtHAP2-AAY51999.1; OsSPCH1-XP_025882342.1, OsSPCH2-XP_015624375.2, OsMUTE-XP_015638702.1, OsFAMA-XP_015638786.1, OsJHAP2-XP_015638244.1; GmSPCH-XP_006596286.1, GmMUTE-XP_003547240.1, GmFAMA-XP_006573288.1, GmHAP2-KAG4987519.1; EgSPCH-XP_010915684.1, EgMUTE-XP_010931955.1, EgFAMA-XP_010909228.1, EgHAP2-XP_010940961.1; PaSPCH-XP_034925606.1; PaMUTE-XP_034930305.1; PaFAMA-XP_034922982.1; PaHAP2-TKR97453.1; EugSPCH-XP_010024801.3, EugMUTE-XP_010024410.3, EugFAMA-XP_018716354.2, EugHAP2-EugHAP2-XP_010053330.2. The phytogenic tree was constructed by using the maximum likelihood method [45] and the JTT matrix-based model [46] with 100 bootstraps via MEGA-X software [47]. 

To identify the signaling pathways of the putative EgSPCH binding targets among DEGs, our DEGs were aligned with the chromatin immunoprecipitation (ChIP) sequencing dataset of AtSPCH targets in *Arabidopsis* [13]. The Gene Ontology (GO) accessions of DEGs were retrieved from the PalmXplore database of oil palm [48]. Gene ontology enrichment analysis and signaling pathway clustering of candidate genes based on the relative expression of DEGs were performed with the program Idep.95 [49] by referencing to *Arabidopsis*. The gene signaling pathway network was analyzed using the BlastKOALA of KEGG [50] using the protein sequence of the putative EgSPCH binding DEGs.

### 4.6. Validation of RNA-Seq Data Using qPCR

The relative expression of *EgSPCH* and 11 randomly selected DEGs was tested by qPCR to examine the validity of the RNA-Seq dataset. The primers used for qPCR are listed in Appendix A. β-tubulin gene was used as a housekeeping gene (internal control) to normalize the relative expression of genes. RT-qPCR was performed in the CFX96 Touch Deep Well Real Time PCR System (Bio-Rad, CA, USA) with the program in a previous study [51]. Each gene for qPCR was performed by a biological/experimental triplicate.

## 5. Conclusions

A proposed regulatory model of *EgSPCH* in response to salt stress was suggested in our study (Figure 7). In brief, salt induces stomatal production of oil palm, which is a result of the activation of the stomatal initiator, EgSPCH. The putative targets of EgSPCH found in different levels of salt stress groups were largely involved in the known biological processes related to salt response [5]. SPCH is functionally conserved but genetically divergent among ligneous and herbaceous plants. In particular, the binding motifs of EgSPCH as a transcription factor are different with *Arabidopsis*, suggesting a different regulatory network of EgSPCH in response to salt stress. The salt response of stomatal production in this study deepened our understanding of the effect of osmotic stress on oil palm and has broad implications for investigating the evolutionary divergence of SPCH in ligneous and herbaceous plants.

## Figures and Tables

**Figure 1 ijms-23-04659-f001:**
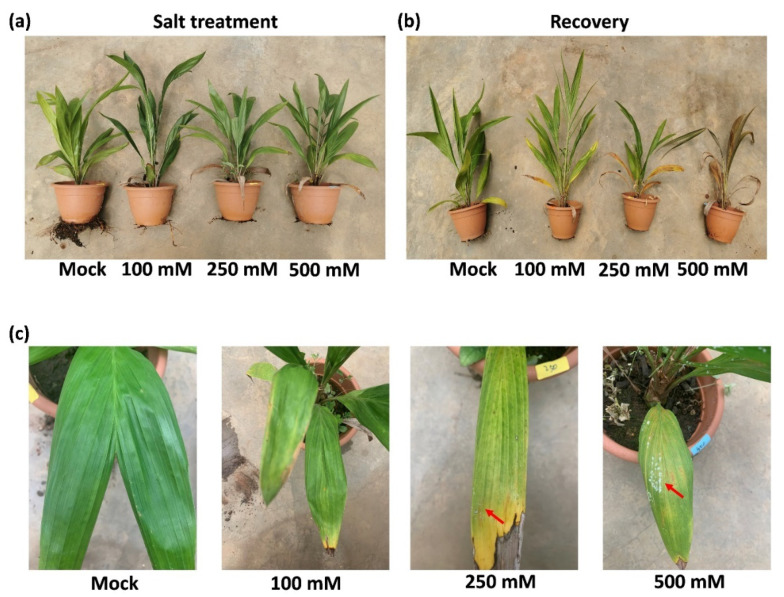
Phenotypical changes of oil palm seedlings in response to different levels of salt stress. (**a**) Circa 2-year-old oil palm seedlings were treated daily with either 150 mL water (Mock) or 150 mL NaCl with four gradient concentration: 100 mM, 250 mM, and 500 mM for 14 days. Four seedlings were used in each group as biological repeats. (**b**) All the seedlings from (**a**) were recovered with 150 mL water daily for another 14 days. (**c**) The leaves of oil palms from (**a**). Red arrows indicate the salt crystals emitted from leaf surface.

**Figure 2 ijms-23-04659-f002:**
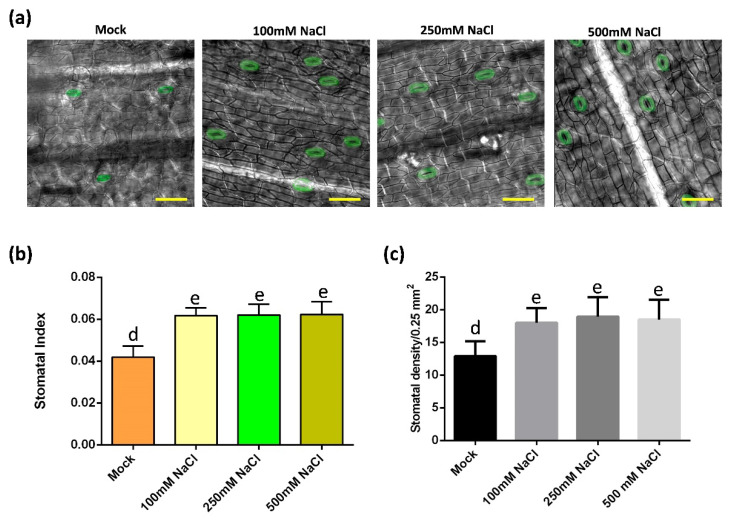
Salt stress activates stomatal production in oil palm. (**a**) Abaxial stomata of oil palm in mock and salt stress groups. Oil palm seedlings were treated with either water or NaCl solutions with 100 mM NaCl, 250 mM NaCl, 500 mM NaCl for 14 days respectively. Stomata are green colored. Scale bar = 30 µm. (**b**) Stomatal index of samples from (**a**). (**c**) Stomatal density of samples from (**a**). Values are mean ± SD; *n* = 20. One-way ANOVA with post-hoc Tukey HSD; *p* < 0.01. Samples were treated daily with 150 mL of either water (Mock) or NaCl for 14 days.

**Figure 3 ijms-23-04659-f003:**
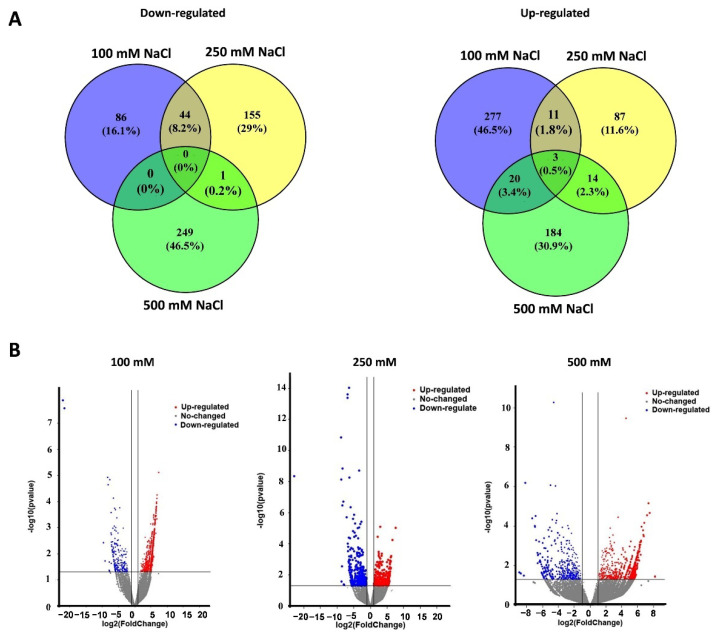
(**A**) The number of up- and down-regulated DEGs in the young leaves of oil palm seedlings under three level of salt stress: 100 mM NaCl, 250 mM NaCl, and 500 mM NaCl; (**B**) the *p*-value and log_2_foldchange (Log_2_FC) of all the genes.

**Figure 4 ijms-23-04659-f004:**
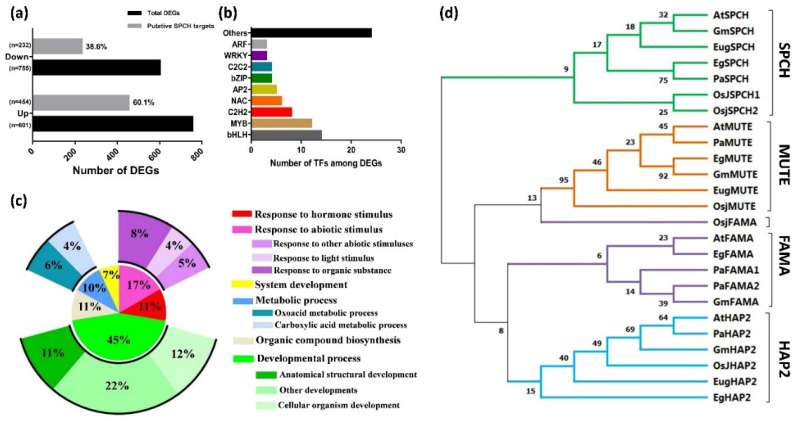
(**a**) Percentage of putative EgSPCH target DEGs in RNA-seq analysis. (**b**) The number of transcription factors among EgSPCH putatively targeted DEGs. (**c**) Enriched GO terms of EgSPCH targets. (**d**) The phylogenic analysis of the protein of SPCH and its two homolog transcription factors, MUTE and FAMA, in *Arabidopsis* (*At*), *japonica rice* (*Osj*), *soybean* (*Gm*), *Elaeis guineensis* (*Eg*), *Eucalyptus grandis* (*Eug*), and *Populus alba L* (*Pa*). Bootstrap = 500. The percentage of replicate trees in which the associated taxa clustered together in the bootstrap test (500 replicates) are shown next to the branches.

**Figure 5 ijms-23-04659-f005:**
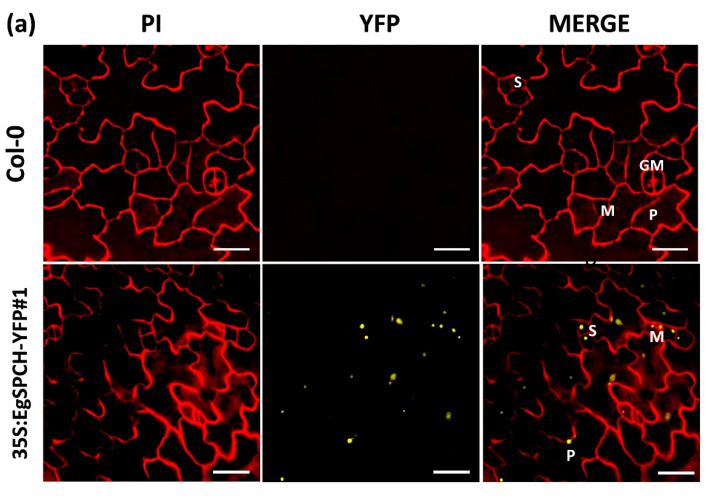
(**a**) Subcellular localization of EgSPCH in the abaxial cotyledons of 3 dpg *Arabidopsis* seedlings. S: stomatal lineage ground cell (SLGC); M: meristemoid; P: pavement cell; GM: guard mother cell. (**b**) The stomatal development of the 7 dpg abaxial cotyledons of Col-0 and the two transgenic *35S:EgSPCH-YFP*(*#1* and *#2*) in *Arabidopsis.* (**c**) The small cell number and pavement cell number of Col-0 and two transgenic 35S:EgSPCH-YFP. Scale bar = 10 μm. Values are mean ± SD; *n* = 20. One-way ANOVA with post-hoc Tukey HSD; *p* < 0.01.

**Figure 6 ijms-23-04659-f006:**
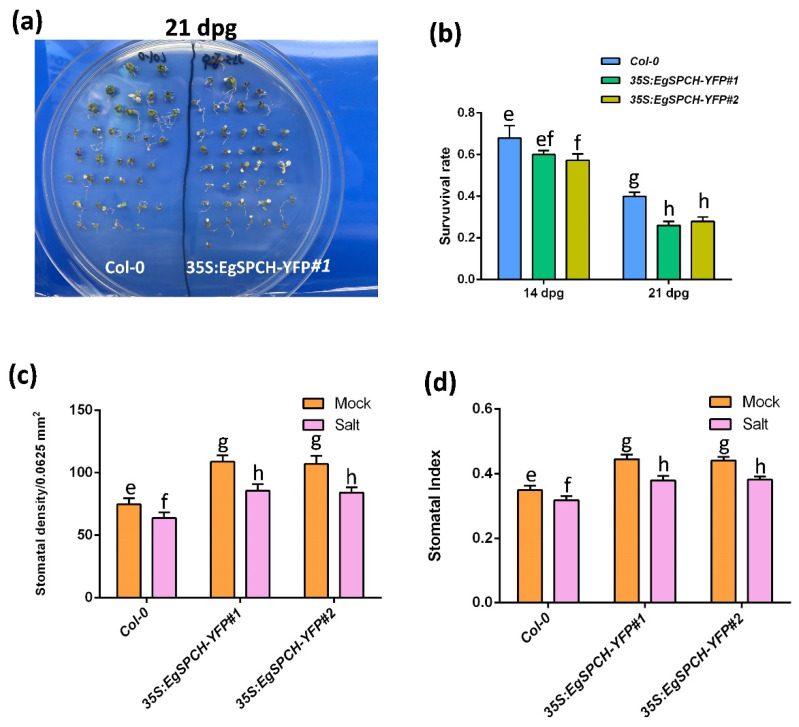
(**a**) Col-0 and *35S-EgSPCH-YFP* seeds were germinated and grown on ½ MS plates for 7 days, after which, they were transferred to ½ MS+ 150 mM NaCl plates for 7 and 14 more days (14 and 21 dpg). (**b**) Survival rate of Col-0 and *35S-EgSPCH-YFP* seedlings from (**a**) at 14 dpg and 21 dpg. *n* = 40. (**c**) Stomatal density of abaxial cotyledons from (**a**) at 14 dpg. (**d**) Stomatal index of samples from (**a**) at 14 dpg. Values are mean ± SD; *n* = 20. One-way ANOVA with post-hoc Tukey HSD; *p* < 0.01.

**Figure 7 ijms-23-04659-f007:**
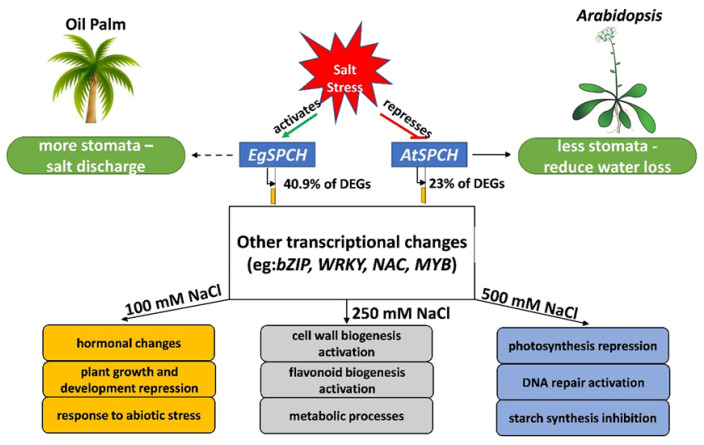
A proposed model of the regulatory networks of EgSPCH in response to different levels of salt stress. In contrast to Arabidopsis, where salt represses the stomatal development to reduce water loss, salt activates the transcription of EgSPCH, which directly increases the stomatal production of oil palm. EgSPCH putatively binds to ~41% of DEGs, including key transcription factors that regulate diverse biological processes. In addition, some GO items, which are involved in the known biological processes in response to salt stress, were found in different concentrations of NaCl.

## Data Availability

Raw RNA-seq reads used in this study have been deposited to the DDBJ DRA database with a DRA submission no. DRA013127.

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
