# Peer review of "EgSPEECHLESS Responses to Salt Stress by Regulating Stomatal Development in Oil Palm"

_ijms, 2022, doi:10.3390/ijms23094659_

Round 1

Reviewer 1 Report

SPCH proteins play important roles in the formation of stomatal in plants. The authors focused on the expression of EgSPCH from the RNA-seq data under the salt treatment conditions in oil palm. Then they postulated the downstream targeted genes by Arabidopsis Chip-seq data and validated the function by transgenic Arabidopsis. However, there are some major concerns as follows.

  1. The present title is not suitable;
  2. The experimental data are needed for validation of some targeted genes of Egspch , such as yeast one hybrid and LUC/REN assays;
  3. The quality of phenotypic photo in Figure 6 a needs further to be improved;
  4. The result of 2,2 should be shown according to Figure 2a,b and c ;
  5. Some English writing need to be improved, such as the DEGs in abstract section, should use the full name for the first time.

Author Response

Dear Reviewer, 

Reviewer 2 Report

Minor Revision

The MS entitled “EgSPEECHLESS and its putative binding targets are involved in the salt response of oil palm” with authors Zhuojun Song, Le Wang, Chong Cheong Lai, Zituo Yang, May Lee and Gen Hua Yue, showed a study of oil palm grown under different levels of salt stress (100, 250 and 500 mM NaCl) for 14d. The authors showed an increase in EgSPCH expression and increased stomatal density of oil palm after salinity. Interestingly, a difference with herbaceous plants was documented after monitoring Arabidopsis. In oil palm, salinity generates more stomata together with increased EgSPCH expression, whereas in Arabidopsis overexpression of EgSPCH increased the stomatal production and lowered the salt tolerance.

The MS need corrections of some faults, so I suggest it be accepted after Minor revision. The necessity of corrections was highlighted within the text.

  • Authors should define the abbreviation (DEG) when it was mentioned for the first time.
  • Authors need to specify which osmolyte exactly do they mean in section 3.1 (above organic osmolyte)
  • If it is not necessary, the authors should remove “well-controlled” from section 4.1.
  • The caption of Figure 5: Authors need to mention if only control plants were used.
  • Supplemental figure S1(e): Probably there is a lapse and Figure 4 should be replaced with Figure 2.

Author Response

Dear Reviewer,

Round 2

Reviewer 1 Report

I think the modified title is still unsuitable .  I suggest the title is 

"EgSPEECHLESS responses to salt stress by regulating stomatal development in oil palm".

Author Response

Dear reviewer,

Thank you again for your assessment and suggestion. We already changed the title suggested by you.